# MVMD-MOMEDA-TEO Model and Its Application in Feature Extraction for Rolling Bearings

**DOI:** 10.3390/e21040331

**Published:** 2019-03-27

**Authors:** Zhuorui Li, Jun Ma, Xiaodong Wang, Jiande Wu

**Affiliations:** Faculty of Information Engineering and Automation, Kunming University of Science and Technology, Kunming 650000, China

**Keywords:** modified variational mode decomposition, multipoint optimal minimum entropy deconvolution adjusted, Teager energy operator demodulation, fault feature extraction, rolling bearings

## Abstract

In order to extract fault features of rolling bearings to characterize their operation state effectively, an improved method, based on modified variational mode decomposition (MVMD) and multipoint optimal minimum entropy deconvolution adjusted (MOMEDA), is proposed. Firstly, the MVMD method is introduced to decompose the vibration signal into intrinsic mode functions (IMFs), and then calculate the energy ratio of each IMF component. The IMF component is selected as the effective component from high energy ratio to low in turn until the total energy proportion *E_sum_*(*t*) ≥ 90%. The IMF effective components are reconstructed to obtain the subsequent analysis signal *x__new_*(*t*). Secondly, the MOMEDA method is introduced to analyze *x__new_*(*t*), extract the fault period impulse component *x__cov_*(*t*), which is submerged by noise, and demodulate the signal *x__cov_*(*t*) by Teager energy operator demodulation (TEO) to calculate Teager energy spectrum. Thirdly, matching the dominant frequency in the spectrum with the fault characteristic frequency of rolling bearings, the fault feature extraction of rolling bearings are completed. Finally, the experiments have compared MVMD-MOEDA-TEO with MVMD-TEO and MOMEDA-TEO based on two different data sets to verify the superiority of the proposed method. The experimental results show that MVMD-MOMEDA-TEO method has better performance than the other two methods, and provides a new solution for condition monitoring and fault diagnosis of rolling bearings.

## 1. Introduction

Rotating machinery is core equipment in commercial production. It is widely used in metallurgy, power, petrochemical, manufacturing, aerospace, and other industrial production fields [1]. Rolling bearing is one of the most frequently used and easily vulnerable key components in rotating machinery. According to incomplete statistics, more than 44% of rotating machinery faults are caused by bearing faults [2]. Therefore, the research on rolling bearing operation condition monitoring and fault diagnosis has important theoretical value and economic significance.

However, the operating conditions of rolling bearings are usually complex and inevitably affected by various noise and signal modulation interference. It is difficult to extract fault characteristics directly from time domain or frequency domain [3]. Therefore, how to extract fault feature information from nonstationary vibration signals is the key to bearing fault diagnosis.

To extract bearing fault features, Hilbert–Huang transform (HHT) [4], kurtosis [5], high-order spectrum [6,7], Wavelet Transform (WT) [8], Empirical Mode Decomposition (EMD) [9], Local Mean Decomposition (LMD) [10], and other methods have been proposed and achieved some results. Nevertheless, these methods have their own limitations. HHT has some problems, such as unexplained negative frequency and energy leakage caused by endpoint effect. High-order spectrum has good application in signal processing and fault feature extraction of nonlinear systems, but its computational complexity is larger than other algorithms. WT needs to preset the wavelet basis and decomposition scale, and the result is a fixed frequency band signal without self-adaptability. Although EMD and LMD methods can adaptively decompose complex signals into a series of components, there are still some theoretical problems, such as envelope, mode aliasing, endpoint effect, and IMF criterion.

Combining the idea of solving modal bandwidth with constrained optimization, Dragomiretskiy and Zosso [11] proposed variational mode decomposition (VMD). This method used an iterative method to search the optimal solution of the variational model, and then determined the central frequency and bandwidth of each component so that the effective separation of signal frequency domain can be realized adaptively. Compared with EMD and LMD, there is no mode mixing and endpoint effect. Because of the above advantages, the VMD method has been widely used in rolling bearing fault feature extraction since it was proposed [12,13,14]. However, there are two limitations for VMD: (1) the number K of decomposition components must be given beforehand and (2) the selection of control parameters for VMD lacks theoretical basis. For nonlinear and unsteady signals, the preset the number K of decomposition modes may lead to information loss or overdecomposition, which affects the performance of subsequent feature extraction [15]. Therefore, how to quickly and adaptively determine the decomposition mode number *K* of VMD for improving the speed of signal processing is one of the urgent problems to be solved. Therefore, the modified variational mode decomposition (MVMD) method proposed in [16] is introduced to determine the decomposition mode number *K* of VMD rapidly and accurately.

A certain amount of noise still remains in each IMF component obtained by decomposition method without exception. To improve the accuracy of fault feature extraction, it is necessary to further enhance the periodic effective pulse of the fault vibration signal such as the bearings, and denoise the reconstructed signal after decomposition. Wiggins [17] proposed minimum entropy deconvolution (MED). H. Endoet et al. firstly applied MED to fault detection of rotating machinery [18]. For reducing the influence of noise and extract the fault feature information of rolling bearings accurately, Sawalhi et al. [19] presented an algorithm for enhancing the surveillance capability of spectral kurtosis by using the minimum entropy deconvolution (MED) technique. The MED technique effectively deconvolved the effect of the transmission path and clarifies the impulses, even where they are not separated in the original signal. Ren et al. [20] proposed a fault characteristics extraction method of rolling bearings based on the combination of VMD and MED. The fault signal of rolling bearing is decomposed by VMD method, and then the reconstructed signal is processed by MED denoising. The fault feature information is extracted from envelope spectrum accurately. However, the MED method is not only complex in operation, but also not necessarily the global optimal filter. Moreover, the MED method is only suitable for single impulse signals. Wang et al. [21] and Xia et al. [22] proposed a bearing fault diagnosis method based on the combination of VMD and maximum correlation kurtosis deconvolution (MCKD). After VMD decomposition of the fault signal, MCKD was used to reduce the noise of each IMF component and highlight the fault impact component to obtain accurate bearing fault characteristic frequency. However, the MCKD method needs to preset the core parameters such as the fault period, which is inconsistent with the reality. Because the fault period may not be known or calculated in advance. To solve the above mentioned problems, McDonald et al. [23] proposed a multipoint optimal minimum entropy deconvolution adjusted (MOMEDA) method, defined the target vector and D-norm, and effectively solved the design problem of the optimal filter. The MOMEDA algorithm does not need to preset the fault cycle, nor does it need to iterate. The impulse component can be accurately extracted by using the multipoint kurtosis spectrum.

In summary, this paper selects the advantages of MVMD and MOMEDA, and proposes a rolling bearing fault feature extraction method based on MVMD and MOMEDA. The advantages of the MVMD-MOMEDA-TEO method are as follows.

(1)The method of MVMD based on scale segmentation is introduced to solve the problem of adaptive selection of mode parameter *K* for VMD decomposition.(2)The introduction of MOMEDA method not only overcomes the limitation of MCKD method, but also highlights the periodic impulse component of bearing fault vibration signal.(3)A new feature extraction method based on MVMD-MOMEDA-TEO is proposed to distinguish the running state of rolling bearings. It provides a new solution for condition monitoring and fault diagnosis of rolling bearings.(4)By using the measured data of four different types of bearing faults from two different sources (Case Western Reserve University (CWRU) and NASA), the comparative experimental analysis of the proposed method and MVMD-TEO method and MOMEDA-TEO method is completed, which validates the effectiveness and feasibility of the proposed method.

The rest of the chapters are arranged as follows. Section 2 describes the basic principles of MVMD and MOMEDA. The implementation details of the presented method are discussed in Section 3. In Section 4, comparative experiments are conducted to demonstrate the effectiveness of the proposed method. Section 5 is discussion and conclusions.

## 2. Core Methodology Introduction

### 2.1. MVMD Method

In the decomposition process, decomposition mode number *K* of VMD needs to be preset, and unreasonable settings can easily lead to information loss or over decomposition. In the literature, a scale space adaptive spectrum segmentation method was introduced. According to the spectrum characteristics of the signal, the support boundary of the signal scale segmentation is selected, and then decomposition mode number *K* in the VMD decomposition process is determined [24]. The adaptive VMD decomposition of the original signal is realized. The method is called MVMD. Its basic supporting principle is briefly described as follows.

#### 2.1.1. VMD

As a new adaptive quasiorthogonal signal decomposition method—VMD—decomposes the given signal *x* into a series of sparse modal components $u_{k}$. Each decomposition component $u_{k}$ has a limited bandwidth of central frequency $w_{k}$.

In order to solve the above mentioned problems, the quadratic penalty factor *α* and Lagrange multiplier operator *λ*(*t*) are introduced to get Equation (1).
(1)$$L(u_{k},\;w_{k},\;\lambda )=\alpha {\sum_{k}\left\|\partial_{t}\left[\left(\delta (t)+\frac{j}{\pi t}\right)*u_{k}(t)\right]e^{-jw_{k}t}\right\|}_{2}^{2}+{\left\|x-\sum_{k}u_{k}\right\|}_{2}^{2}+\langle \lambda ,\;x-\sum_{k}u_{k}\rangle$$

The specific implementation steps are as shown in Figure 1.

#### 2.1.2. Scale Space Representation

Discrete Fourier transform transforms the time domain sampling of discrete vibration signals into frequency domain sampling. Scale space representation can describe the spectrum of signals from different dimensions. So the implementation of scale space representation is shown in Figure 2. 

The derivation process of specific parameters is detailed in [16]. In Figure 1, *n* is a scale parameter. The selection of scale parameters is generally calculated according to Equation (2):(2)$$\sqrt{n}\geq \mu f_{ch}$$

In Equation (2), there is no strict restriction on the selection of μ values, and the recommended range of values is [2,3,4]. At the same time, when the signal is modulated by fault characteristic frequency *f_ch_* and noise pollution, the small change of scale parameters has no obvious influence on the final analysis results. Therefore, the scale parameter is chosen as $\sqrt{n}\geq 3f_{ch}$ [16].

### 2.2. MOMEDA Method

The purpose of the MOMEDA algorithm is to find the optimal finite impulse response (FIR) filter in a noniterative way and reconstruct the vibration and shock signal *y*. The deconvolution process is as follows
(3)$$y=f*x=\sum_{k=1}^{N-L}f_{k}x_{k+L-1}$$
k=1,2,⋯,N−L, according to the characteristics of periodic impulse signal, the method introduces multipoint D-norm:(4)$$MDN(y,t)=\frac{1}{\left||t|\right|}\frac{t^{T}y}{\left||y|\right|}$$
(5)$$MOMEDA=\underset{f}{\max}MDN(y,t)=\underset{f}{\max}\frac{t^{T}y}{\left||y|\right|}$$

In the Equation (4), the constant vector *t* is used to determine the position and weight of the target impact component. The optimal filter *f* is obtained by solving the maximum of the multipoint D-norm, and the deconvolution process also obtains the optimal solution.

MOMEDA uses multipoint kurtosis (MKurt) to determine the maximum position of the pulse.
(6)$$MKurt=(\sum_{n=1}^{N-L}t_{n}{}^{2}{)}^{2}\sum_{n=1}^{N-L}{\left(t_{n}y_{n}\right)}^{4}/(\sum_{n=1}^{N-L}t_{n}{}^{8}(\sum_{n=1}^{N-L}y_{n}^{2}{)}^{2})$$

Referring to the parameter selection rule in reference [25], the length range of the filter is 20–500, and the periodic parameters should cover the frequency range analyzed. This article takes the default value of T=[10:0.1:300], L=500 and carries on the analysis.

### 2.3. TEO Demodulation Principle

For continuous signal *x*(*t*): the definition of TEO demodulation *φ*[*x*(*t*)] can be referred to [26]:(7)$$\varphi [x(t)]\;=\;[\dot{x}(t){]}^{2}\;-\;x(t)\ddot{x}(t)$$
$\dot{x}(t)$ and $\ddot{x}(t)$ are the first- and second-order differentials of *x*(*t*) to time t, respectively.

For discrete signal *x*(*n*), *φ*[*x*(*n*)] is defined as
*φ*[*x*(*n*)] = [*x*(*n*)]^2^ − *x*(*n* − 1)*x*(*n* + 1)(8)

It is known from Equation (8) that for discrete signal *x*(*n*) TEO only needs three sets of sample data to calculate the signal source energy at any time *n*. For the IMF component of the vibration signal, the TEO demodulation envelope signal *φ*[*PF*] of the IMF component can be calculated according to Equation (8), and the subsequent Fourier spectrum analysis is performed using *φ*[*PF*] instead of the original signal *x*(*n*). The spectral characteristics of the vibration signal are extracted to determine the fault. 

## 3. MVMD-MOMEDA-TEO Implementation Process

Combining the advantages of MVMD and MOMEDA with TEO, the implementation flow chart is shown in Figure 3. The detailed implementation steps of the proposed method are described as follows.

Step 1: Calculate the fault characteristic frequency *f_ch_*, and then obtain the scale parameters $\sqrt{n}$.

Step 2: Collect the rolling bearings vibration signal *x*(*t*) and calculate scale space representation *L*(*f*, *n*) of the Fourier spectrum. Then, the support boundary number *m* of the signal spectrum is acquired. Finally, the decomposition modes number K=m is decided in VMD based on the support boundary.

Step 3: Penalty parameter *α* = 2000, VMD bandwidth *τ* = 0.001.

Step 4: According to the given parameters of Step 2 and Step 3, the signal *x*(*t*) is decomposed by VMD into a series of IMF components *u_k_*(*t*).

Step 5: According to the energy ratio criterion (on the basis of the energy ratio from high-to-low, the IMF component is selected as the effective component in turn until the total energy proportion *E_sum_*(*t*) ≥ 90%), reconstruct the analysis signal *x__new_*(*t*).

Step 6: The MOMEDA method is used to deconvolute *x__new_*(*t*) to suppress the influence of noise interference, enhance the periodic impulse component of *x__new_*(*t*), and finally obtain the deconvolution fault characteristic signal *x__cov_*(*t*).

Step 7: Teager energy spectrum of *x__cov_*(*t*) is calculated by Teager energy operator, and finally the fault feature is extracted.

## 4. Comparative Analysis of Experiments

In this paper, the comparative experiments of MVMD-MOMEDA-TEO, MVMD-TEO, and MOMEDA-TEO are completed by using CWRU data sets [27] and NASA data sets [28]. The effectiveness and superiority of the proposed method are further verified by experimental analysis.

### 4.1. CWRU Rolling Bearing Vibration Data Analysis

The CWRU rolling bearing fault analogous experimental platform and the actual bearing are shown in Figure 4, and the detailed experimental parameters are shown in Table 1. The speed of the motor is 1797 rpm (i.e., the rotation frequency *f_r_* = 1797/60 Hz = 29.95 Hz). The sampling frequency *f_s_* is 12 kHz and the data point *N* is 2048. 

Outer race defect frequency:(9)$$BPFO=\frac{Z}{2}\left(1-\frac{d}{D}\cos\theta \right)\times f_{r}$$

Inner race defect frequency:(10)$$BPFI=\frac{Z}{2}\left(1+\frac{d}{D}\cos\theta \right)\times f_{r}$$

Based on the bearing parameters shown in Table 1 and Equations (9) and (10), the fault characteristic frequency of rolling bearing, BPFO = 107.36 Hz, and BPFI = 162.19 Hz, are calculated, respectively.

In the following sections, the comparative experiments of the two methods are completed by using the vibration signals of the outer race fault and the inner race fault in CWRU, respectively.

#### 4.1.1. The Feature Extraction of Outer Race Fault

Figure 5 and Figure 6 show the results of time-domain and frequency-domain waveforms in normal operation and with an outer race fault, respectively. It can be seen from Figure 5a and Figure 6a that with the continuous operation of bearings, the time-domain waveforms of vibration signals have obvious impulse components with certain regularity, and there are many unknown components in the spectrum. Therefore, more noise can be observed. However, from Figure 5b and Figure 6b, it cannot directly obtain the detailed fault information, such as fault type, fault location, etc. It is necessary to adopt new analytical solutions or methods to extract the characteristic frequency of rolling bearing and distinguish the running state of rolling bearings. For this perspective, follow-up analysis is carried out by using MVMD-MOMEDA-TEO and MOMEDA-TEO.

##### Experiment of Outer Race Fault Feature Extraction Based on the MVMD-MOMEDA-TEO Method

According to Figure 7, 10 IMF components are derived by VMD method. The energy ratio *E*(*t*) of each IMF component is calculated separately and shown in Table 2. According to the energy ratio from high-to-low, the IMF component is selected as the effective component in turn until the total energy proportion *E_sum_*(*t*) ≥ 90%. Based on this criterion, five IMF components (IMF4–IMF8) are selected and shown in Figure 8.

Figure 9a is the reconstruction signal *x__new_* (*t*) composed by five selected components. Next, the periodic pulse signal *x__cov_*(*t*) is extracted from *x__new_* (*t*) by the MOMEDA method and is demonstrated in Figure 9b. Finally, the Teager energy spectrum is calculated by Teager Energy Operator demodulation and displayed in Figure 10. It can be seen that frequency 105.5 Hz and its frequency doubling characteristic approach theoretical BPFO and its frequency doubling (2BPFO ~ 9BPFO). It can be judged that the outer race fault has occurred.

##### Experiment of Outer Race Fault Feature Extraction Based on MOMEDA-TEO Method

Figure 11 shows that the original signal *x*(*t*) is denoised by MOMEDA filtering directly, and *x__cov_*(*t*) is obtained. It can be seen that frequency 105.5 Hz and its frequency doubling characteristic approach BPFO and its frequency doubling (2BPFO~9BPFO) in Figure 12. So it can seen that the outer race fault has occurred.

The comparative experiments of MVMD-MOMEDA-TEO, MVMD-TEO in Ref. [16], and MOMEDA-TEO show that (1) the proposed MVMD-MOMEDA-TEO can achieve the comparable or better results than the other two methods, and clearly identify the characteristic frequency BPFO and multiple harmonics of outer race fault and (2) at the same time, it can be clearly observed that MVMD-MOMEDA-TEO and MOMEEDA-TEO can achieve better performance than MVMD-TEO, and their Teager energy spectrum amplitude is more obvious and prominent. Therefore, the impact part of vibration signal of rolling bearing fault can be enhanced by MOMEDA in actual analysis and the necessity of MOMEDA deconvolution is also demonstrated.

#### 4.1.2. The Feature Extraction of Inner Race Fault

Figure 13 and Figure 14 show the results of time-domain and frequency-domain waveforms of normal operation and inner race fault, respectively. The presence of more noise can be observed from Figure 13a and Figure 14a. However, from Figure 13b and Figure 14b, it cannot directly obtain the detailed fault information. So the following analysis was carried out by using MVMD-MOMEDA-TEO and MOMEDA-TEO.

##### Experiment of Inner Race Fault Feature Extraction Based on the MVMD-MOMEDA-TEO Method

According to Figure 15, 11 IMF components are derived by the VMD method. The energy ratio *E*(*t*) is calculated and shown in Table 3. According to the energy ratio from high-to-low, the IMF component is selected as the effective component in turn until the total energy proportion *E_sum_*(*t*) ≥ 90%. Based on this criterion, seven IMF components (IMF2–IMF3 and IMF5–9) were selected and shown in Figure 16.

Figure 17a is *x__new_* (*t*) by seven selected components. Next, *x__cov_*(*t*) is extracted from *x__new_* (*t*) by the MOMEDA method and shown in Figure 17b. Finally, the Teager energy spectrum is calculated by TEO and shown in Figure 18. It can be seen that characteristic frequency 164.10 Hz, its frequency doubling approach BPFI, and its frequency doubling (2BPFI~6BPFI). From this, it can be seen that the inner race fault has occurred in the rolling bearings.

##### Experiment of Inner Race Fault Feature Extraction Based on MOMEDA-TEO Method

Figure 19 shows that *x*(*t*) is denoised by MOMEDA filtering directly, and *x__cov_*(*t*) is obtained. It can be seen that frequency 164.10 Hz and its frequency doubling approach BPFI and its frequency doubling (2BPFI~6BPFI) in Figure 20. From this, it can be seen that the inner race fault has occurred.

Through the analysis of vibration signal of inner race fault, a conclusion which is basically consistent with outer race fault can be obtained. However, some hidden phenomena revealed that the performance of the inner race fault is slightly worse than outer race fault in depth, which may be relevant to the impact of inner race parameter error and transmit process.

### 4.2. NASA Rolling Bearing Vibration Data Analysis

The comparative experiments among MVMD-MOEDA-TEO, MVMD-TEO, and MOEDA-TEO are completed by using the vibration data of rolling bearings with two different fault types of CWRU. The advantages of the proposed method are preliminarily verified, and the experimental results of the proposed method are extended and applicable. Three groups of experiments are completed by using vibration data of rolling bearings from two different fault types of NASA. Figure 21 shows the simulation test platform and sensor layout of the rolling bearing fault. Four bearings are installed on the rotating axle of the test bench. One acceleration sensor is installed on the axial and radial direction of each bearing, and the sampling frequency is 20 kHz. The rotational speed of the motor is 2000 rpm (i.e., the rotational frequency *f_r_* = 2000/60 Hz = 33.33 Hz). Detailed experimental parameters of the bearing are shown in Table 4 [16].

Based on the bearing parameters shown in Table 4 and Equations (9) and (10), the fault characteristic frequencies of rolling bearing, BPFO = 236.4 Hz and BPFI = 296.93 Hz, are calculated, respectively. 

#### 4.2.1. The Feature Extraction of Outer Race Fault

Figure 22 and Figure 23 show the results of time-domain and frequency-domain waveforms of normal operation and the outer ring fault, respectively. It can be observed that more noise occurs, as shown in Figure 22a and Figure 23a. However, from Figure 22b and Figure 23b, it cannot directly obtain the detailed fault information, such as fault type, fault location, etc. So the following analysis was carried out by using MVMD-MOMEDA-TEO and MOMEDA-TEO.

##### Experiment of Outer Race Fault Feature Extraction Based on MVMD-MOMEDA-TEO Method

According to Figure 24, 12 IMF components are derived by VMD method. The energy ratio *E*(*t*) *t* is calculated and shown in Table 5. According to the energy ratio from high-to-low, the IMF component is selected as the effective component in turn until the total energy proportion *E_sum_*(*t*) ≥ 90%. Based on this criterion, six IMF components (IMF2 and IMF4–IMF8) are selected and shown in Figure 25.

Figure 26a is *x__new_* (*t*) by six selected components. Next, *x__cov_*(*t*) is extracted from *x__new_* (*t*) by the MOMEDA method and demonstrated in Figure 26b. Finally, the Teager energy spectrum is calculated by TEO displayed in Figure 27. It can be seen that characteristic frequency 230.70 Hz and its frequency doubling characteristic approach BPFO and its frequency doubling (2BPFO~4BPFO). From this, it can be judged that the failure of outer race of rolling bearing has occurred.

##### Experiment of Outer Race Fault Feature Extraction Based on MOMEDA-TEO Method

Figure 28 shows that *x*(*t*) is denoised by MOMEDA filtering directly, and *x__cov_*(*t*) is derived. Its characteristic frequency 230.70 Hz, frequency doubling approach BPFO, and frequency doubling (2BPFO ~ 4BPFO) are shown in Figure 29. So it can be seen that the outer race fault has occurred.

The comparative experiments of MVMD-MOMEDA-TEO, MVMD-TEO in Ref. [16], and MOMEDA-TEO show that the proposed MVMD-MOMEDA-TEO can achieve the same results as the other two methods, and can clearly identify the outer race fault characteristic frequency and its frequency doubling characteristics.

#### 4.2.2. The Feature Extraction of Inner Race Fault

Figure 30 and Figure 31 show the results of time-domain and frequency-domain waveforms of normal operation and inner race fault, respectively; more noise can be observed, as shown in Figure 30a and Figure 31a. However, from Figure 30b and Figure 31b, it cannot directly obtain the detailed fault information. The following analysis is carried out by using MVMD-MOMEDA-TEO and MOMEDA-TEO.

##### Experiment of Inner Race Fault Feature Extraction Based on MVMD-MOMEDA-TEO Method

According to Figure 32, 12 IMF components are derived by VMD method. The energy ratio *E*(*t*) is calculated and shown in Table 6. According to the energy ratio from high-to-low, the IMF component is selected as the effective component in turn until the total energy proportion *E_sum_*(*t*) ≥ 90%. Based on this criterion, six IMF components (IMF12 and IMF1–IMF8) are selected and shown in Figure 33.

The analysis signal *x__new_* (*t*) is reconstructed by nine components and *x__cov_*(*t*) is extracted from *x__new_* (*t*) by MOMEDA. Finally, the Teager Energy Operator Demodulation of *x__cov_*(*t*) is carried out, and the Teager energy spectrum is calculated as shown in Figure 34. It can be seen that there is obvious peak value at 293.6 Hz in the Teager energy spectrum, which gets close to theoretical frequency of bearing inner race fault, and peak values also occur at 148.39 Hz (0.5 octave) and 439.9 Hz (1.5 octave). From this, it can be judged that the inner race fault occurred in the bearing, which is consistent with the practical fault. However, its characteristic frequency is not obvious in the inner loop.

##### Experiment of Inner Race Fault Feature Extraction Based on MOMEDA-TEO Method

Figure 35 shows that the original signal *x*(*t*) is denoised by MOMEDA filtering directly, and the periodic pulse signal *x__cov_*(*t*) is obtained. Finally, the output signal is demodulated and analyzed by TEO, and its Teager energy spectrum is shown in Figure 36. It can be seen that there is obvious peak value at 293.6 Hz in Teager energy spectrum, which approximates theoretical frequency of bearing inner race fault. From this, it can be judged that the inner race fault occurred in the bearing, which is close to the practical fault. 

Through the comparative experiments of MVMD-MOMEDA-TEO, MVMD-TEO in Ref. [16], and MOMEDA-TEO, we can see that the proposed MVMD-MOMEDA-TEO can achieve the same results as the other two methods, and can clearly identify the inner race fault characteristic frequency and its frequency doubling features. Compared with the results of the other two methods, there are a lot of noise signals in the Teager energy spectrum when using the MVMD method, which affect the extraction of fault features. Thus, it is effective to enhance the impact part of signals by using MOMEDA method. 

## 5. Discussion and Conclusions 

### 5.1. Discussion

An improved method based on MVMD and MOMEDA to extraction fault characteristics for rolling bearings is proposed.

(1)The proposed MVMD-MOMEDA-TEO can achieve the same results as the other two methods, and can clearly identify the fault characteristic frequency (including frequency doubling features) of rolling bearings. (2)Compared with the other two methods, the results obtained by MVMD directly in Ref. [16] have a large number of noise signals in the Teager energy spectrum, which have a certain impact on fault feature extraction. Thus, MOMEDA of complex signals can enhance the impact part of signals. (3)Because the measured signal of NASA inner race contains not only noise interference, but also harmonic signal interference from the outer race and rolling body; the results obtained by MOMEDA method directly are compared with those of other two methods. Due to the signal decomposition process is not carried out, there is a problem that useful fault information will be filtered out together when filtering and denoising. Therefore, it is necessary to decompose the complex signal by MVMD method to extract useful information from the original signal. (4)By comparing the experimental results of the two groups of measured data, it can be seen that the MVMD-MOMEDA-TEO method can get a bit better or equivalent experimental results than the other two methods under strong noise interference, which proves the validity of the proposed method.

### 5.2. Conclusions

A method based on MVMD and MOMEDA is proposed. The following three conclusions are drawn.

(1)Introducing the MVMD method self-adaptively choosing VMD decomposition mode number to realize fast self-adapting decomposition of signals. At the same time, introducing energy proportion index to extract effective decomposition components and reduce signal interference components. (2)The MOMEDA method is introduced to enhance the fault periodic pulse characteristics, and the Teager energy operator is introduced to analyze the envelope demodulation of deconvolution signal *x__cov_*(*t*), which enhances the fault characteristic frequency of rolling bearings in the envelope spectrum. (3)Based on the vibration data of four different fault types from two different datasets of CWRU and NASA, the comparative experiments of MVMD-MOMEDA-TEO and MVMD-TEO, MOMEDA-TEO were carried out systematically, and the validity of the proposed method was demonstrated. 

When the MOMEDA method was used, there are randomness and trial-and-error in the parameter settings of the filter. How to find the best parameter adaptively is an important breakthrough point in improving MOMEDA method.

## Figures and Tables

**Figure 1 entropy-21-00331-f001:**
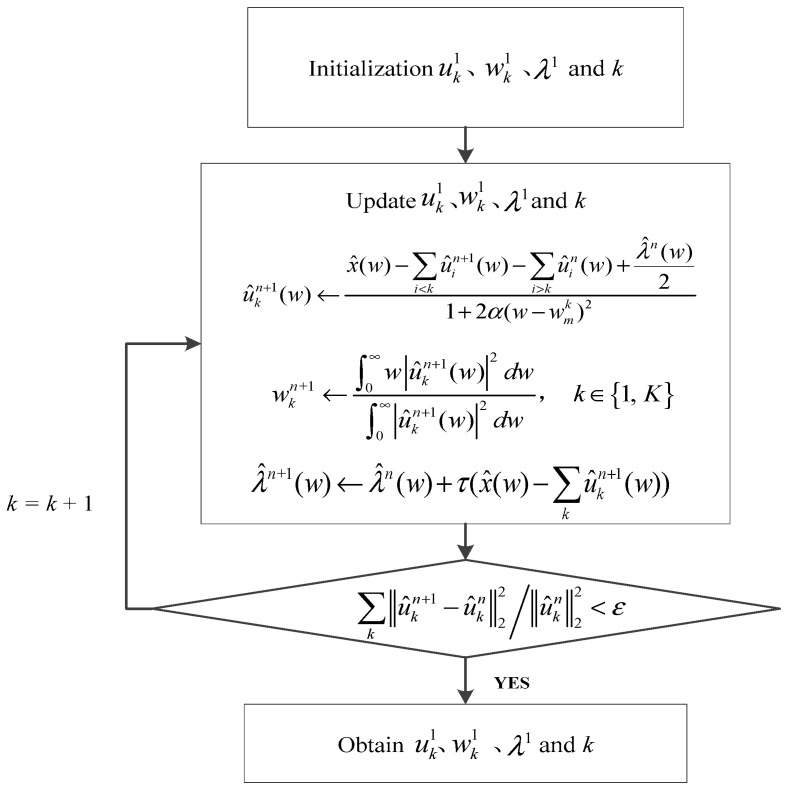
Variational mode distribution (VMD) implementation process.

**Figure 2 entropy-21-00331-f002:**
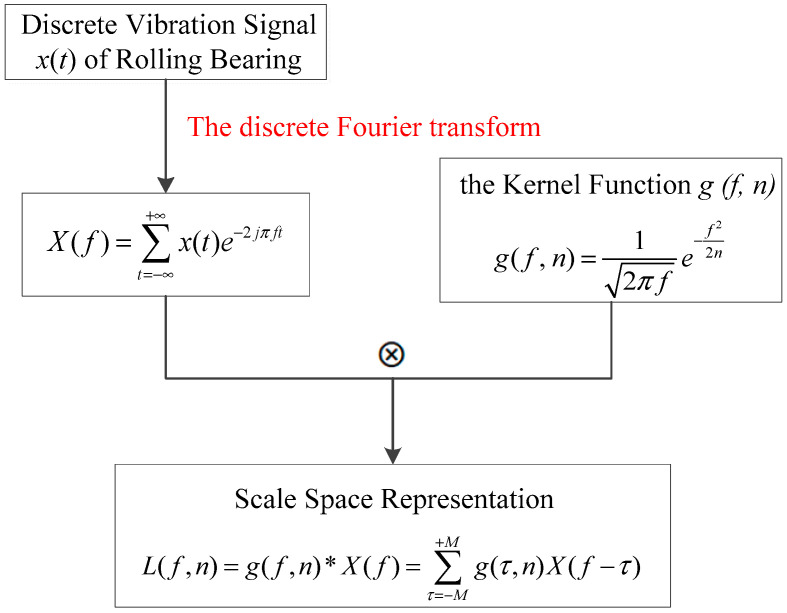
Scale space representation process.

**Figure 3 entropy-21-00331-f003:**
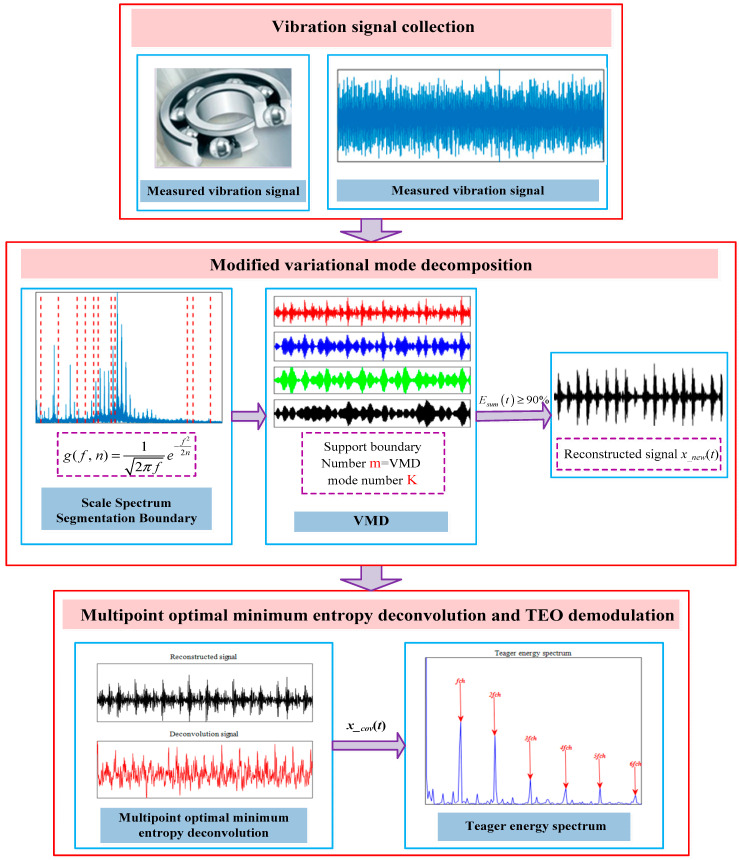
Flow diagram of MVMD-MOMEDA-TEO method implementation.

**Figure 4 entropy-21-00331-f004:**
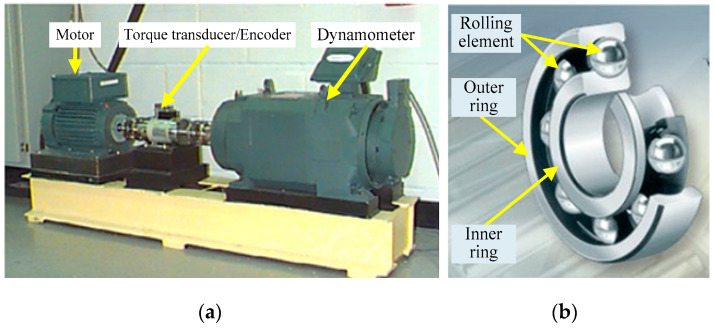
Rolling bearing fault simulation experimental platform and experimental bearing. (**a**) Rolling bearing fault simulation experiment platform. (**b**) Deep groove rolling bearing.

**Figure 5 entropy-21-00331-f005:**
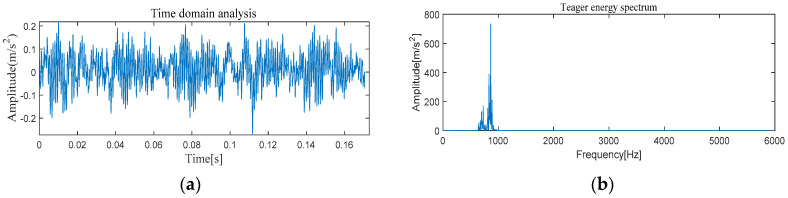
Original normal signal time–frequency analysis. (**a**) Time domain analysis. (**b**) Frequency domain analysis.

**Figure 6 entropy-21-00331-f006:**
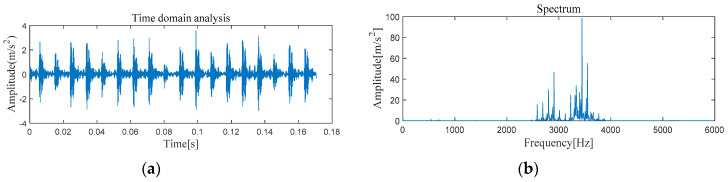
Original fault signal time–frequency analysis. (**a**) Time domain analysis. (**b**) Frequency domain analysis.

**Figure 7 entropy-21-00331-f007:**
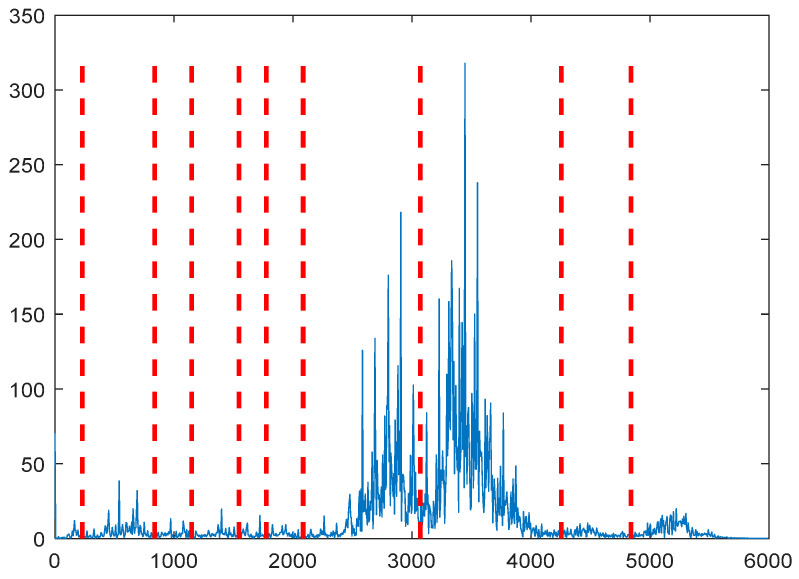
Scale spectrum segmentation boundary.

**Figure 8 entropy-21-00331-f008:**
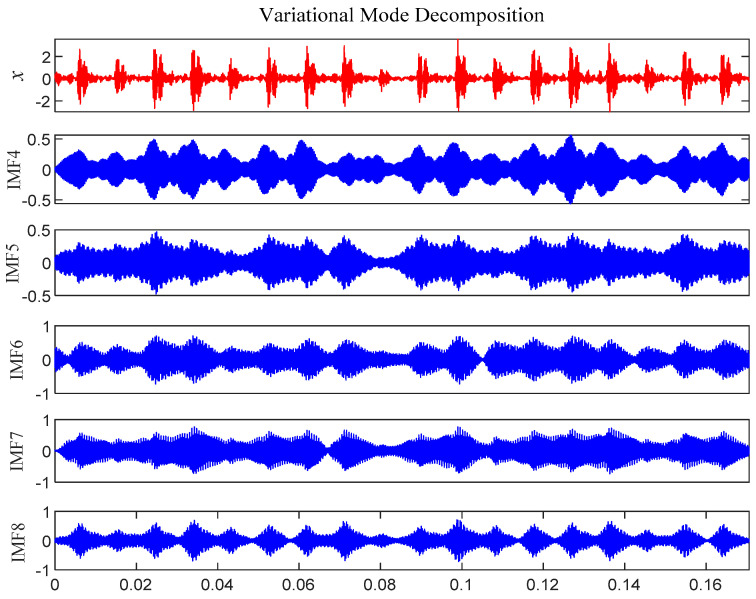
Intrinsic mode function (IMF) effective component time domain waveform.

**Figure 9 entropy-21-00331-f009:**
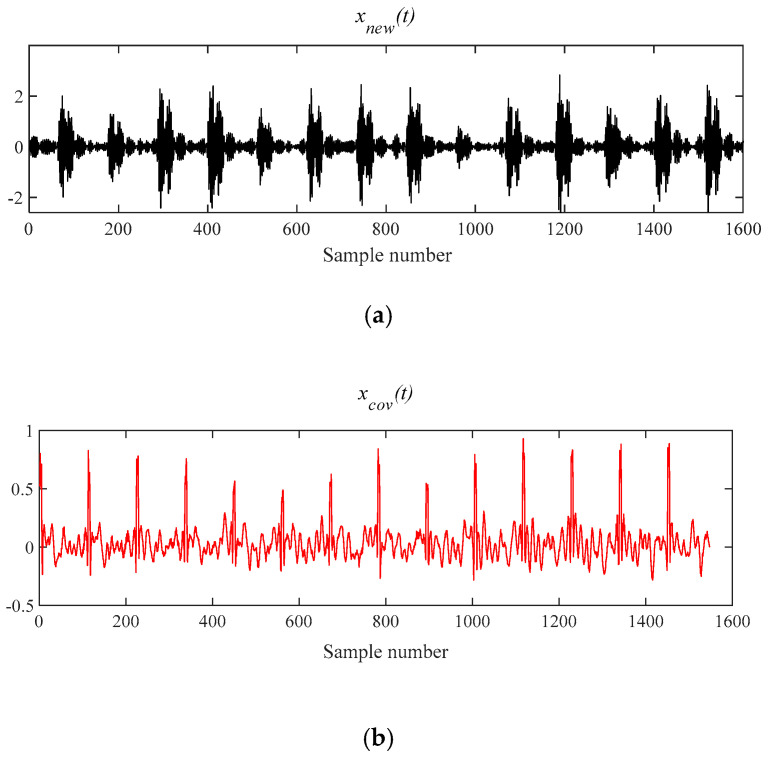
The signal processing by MOMEDA method. (**a**) Reconstruction signal *x__new_* (*t*); (**b**) Periodic pulse signal *x__cov_*(*t*).

**Figure 10 entropy-21-00331-f010:**
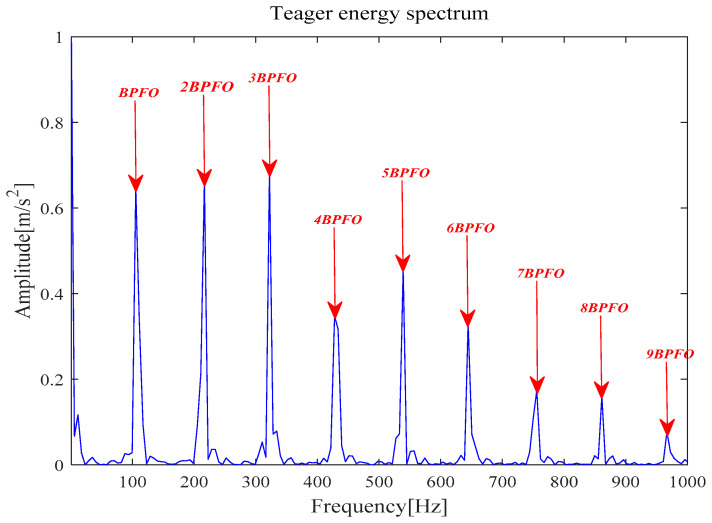
Teager energy spectrum.

**Figure 11 entropy-21-00331-f011:**
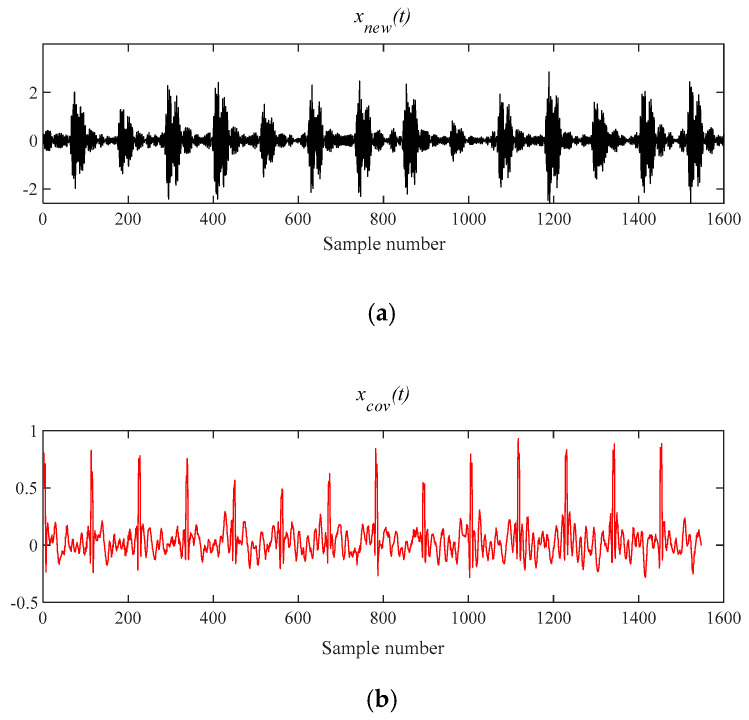
The signal processing by MOMEDA method. (**a**) Reconstruction signal *x__new_* (*t*); (**b**) Periodic pulse signal *x__cov_*(*t*).

**Figure 12 entropy-21-00331-f012:**
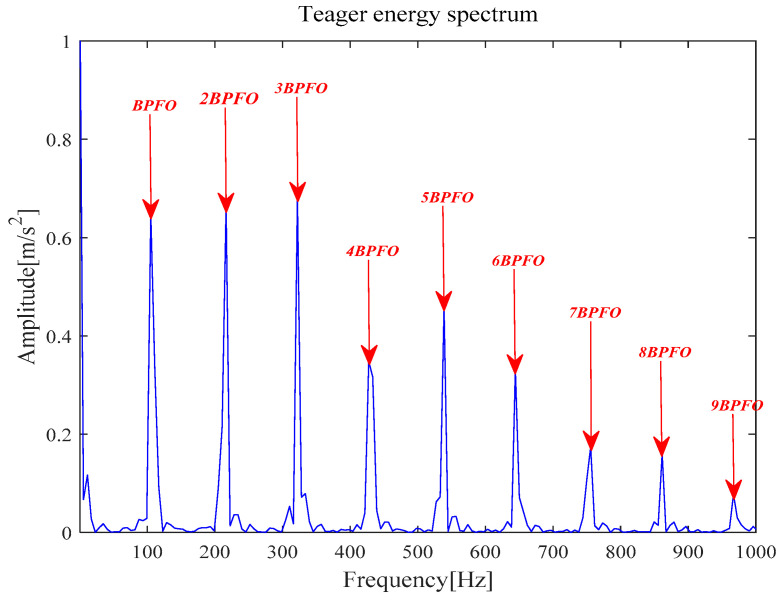
Teager energy spectrum.

**Figure 13 entropy-21-00331-f013:**
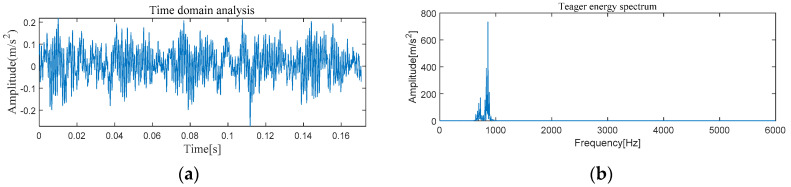
Original normal signal time–frequency analysis. (**a**) Time domain analysis. (**b**) Frequency domain analysis.

**Figure 14 entropy-21-00331-f014:**
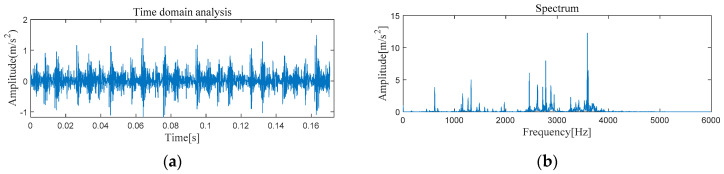
Original fault signal time–frequency analysis. (**a**) Time domain analysis. (**b**) Frequency domain analysis.

**Figure 15 entropy-21-00331-f015:**
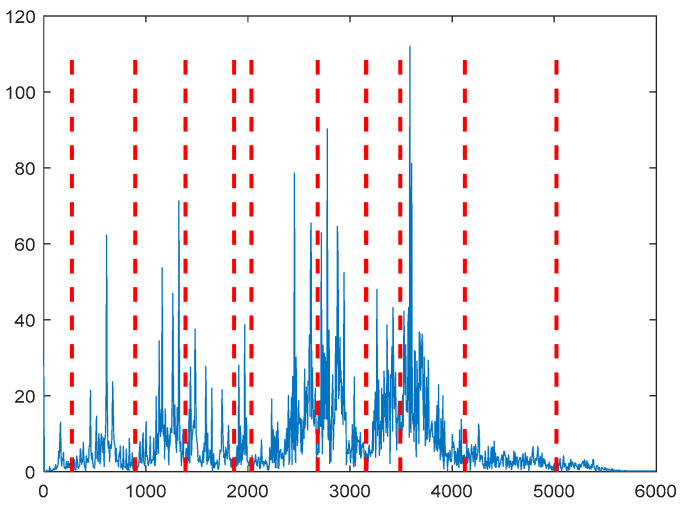
Scale spectrum segmentation boundary.

**Figure 16 entropy-21-00331-f016:**
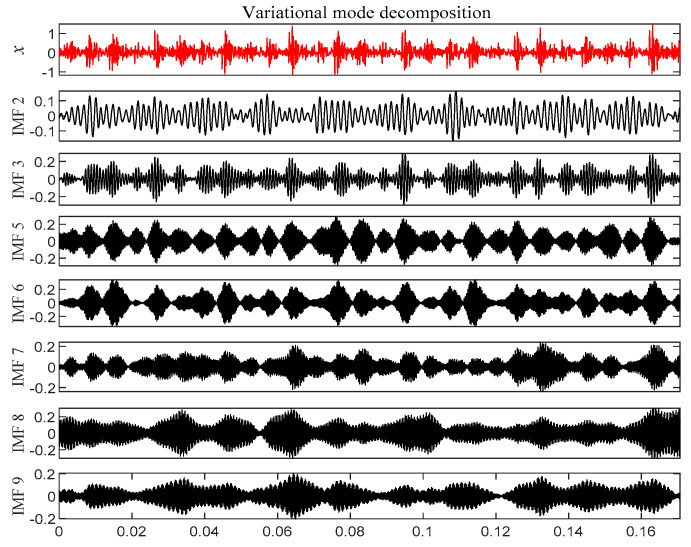
IMF effective component time domain waveform.

**Figure 17 entropy-21-00331-f017:**
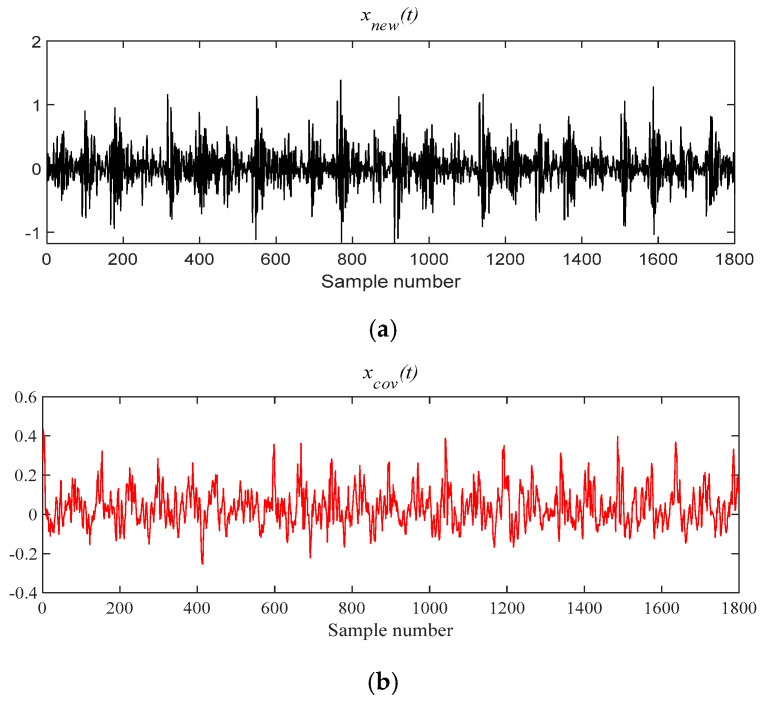
The signal processing by MOMEDA method. (**a**) Reconstruction signal *x__new_* (*t*); (**b**) Periodic pulse signal *x__cov_*(*t*).

**Figure 18 entropy-21-00331-f018:**
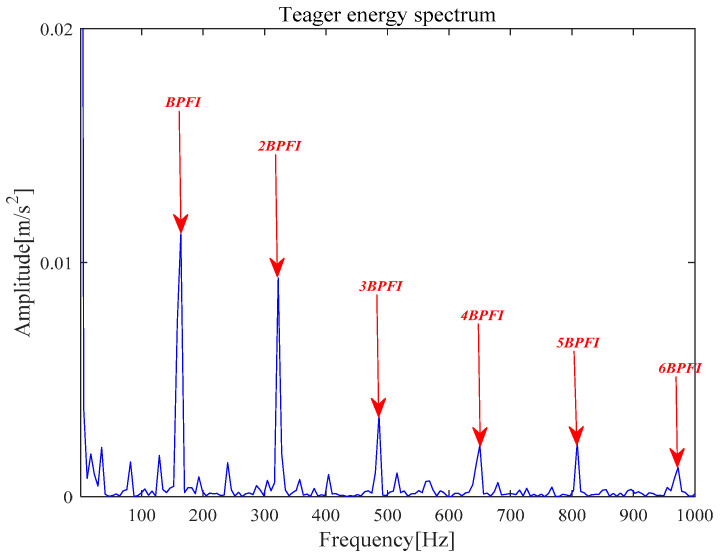
Teager energy spectrum.

**Figure 19 entropy-21-00331-f019:**
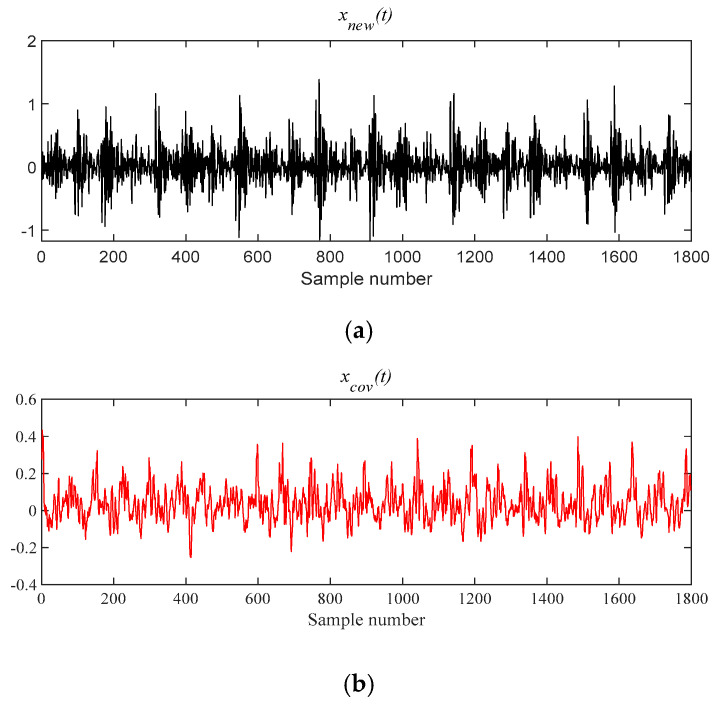
The signal processing by MOMEDA method. (**a**) Reconstruction signal *x__new_* (*t*); (**b**) Periodic pulse signal *x__cov_*(*t*).

**Figure 20 entropy-21-00331-f020:**
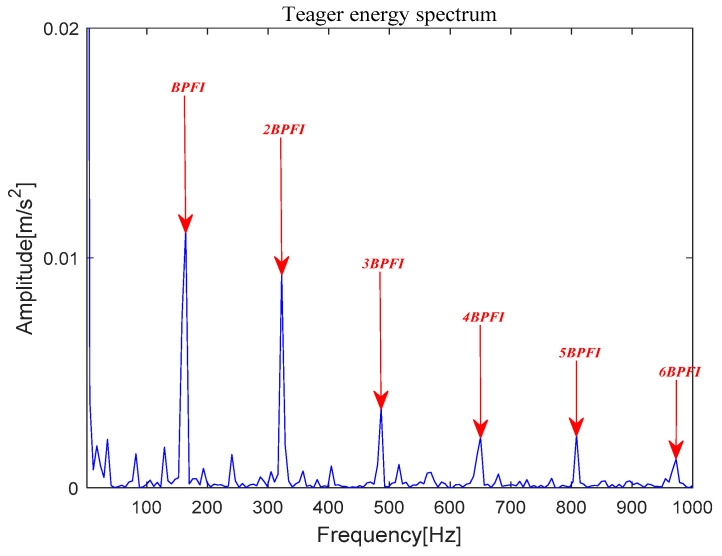
Teager energy spectrum.

**Figure 21 entropy-21-00331-f021:**
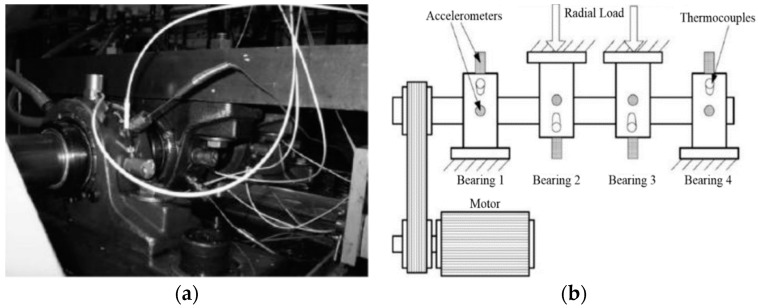
Rolling bearing fault simulation experimental platform and experimental bearing. (**a**) Test bench. (**b**) Sensor layout.

**Figure 22 entropy-21-00331-f022:**
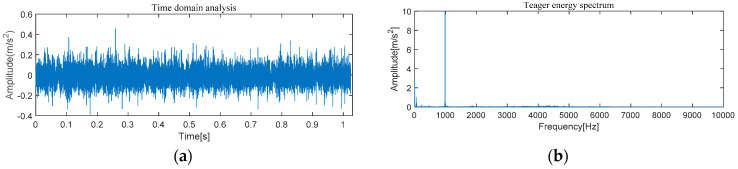
Original normal signal time–frequency analysis. (**a**) Time domain analysis. (**b**) Frequency domain analysis.

**Figure 23 entropy-21-00331-f023:**
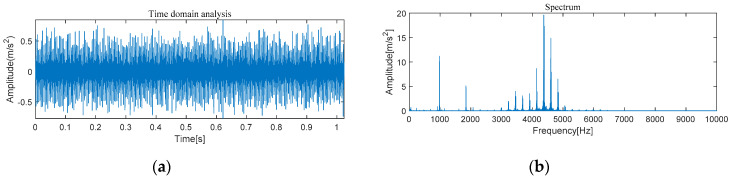
Original fault signal time–frequency analysis. (**a**) Time domain analysis. (**b**) Frequency domain analysis.

**Figure 24 entropy-21-00331-f024:**
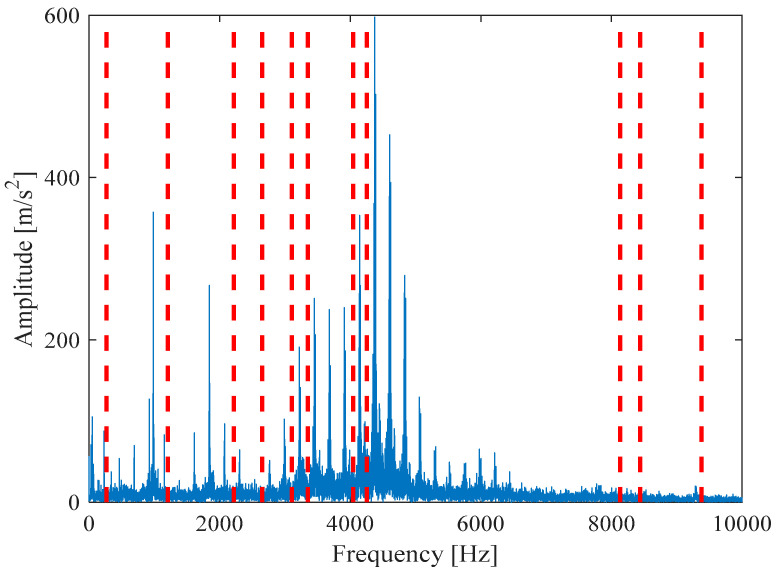
Scale spectrum segmentation boundary.

**Figure 25 entropy-21-00331-f025:**
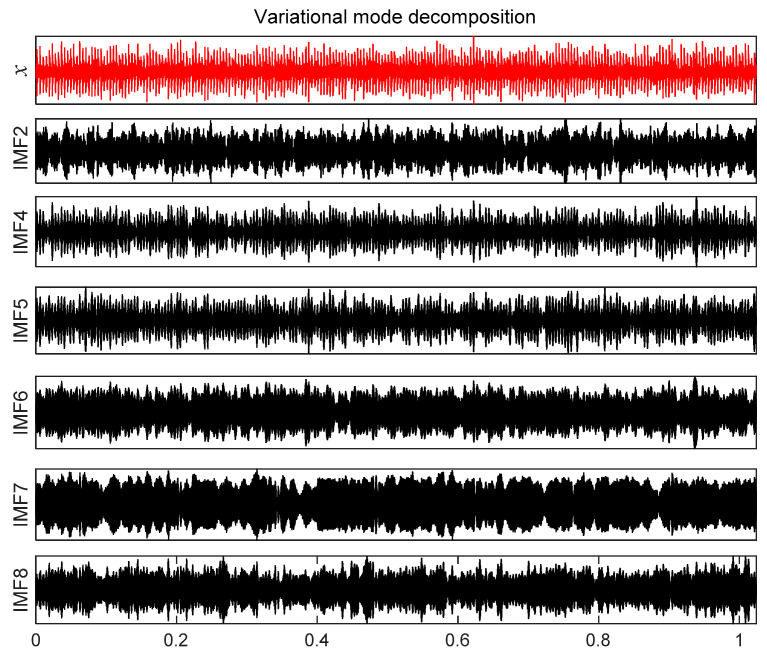
IMF effective component time domain waveform.

**Figure 26 entropy-21-00331-f026:**
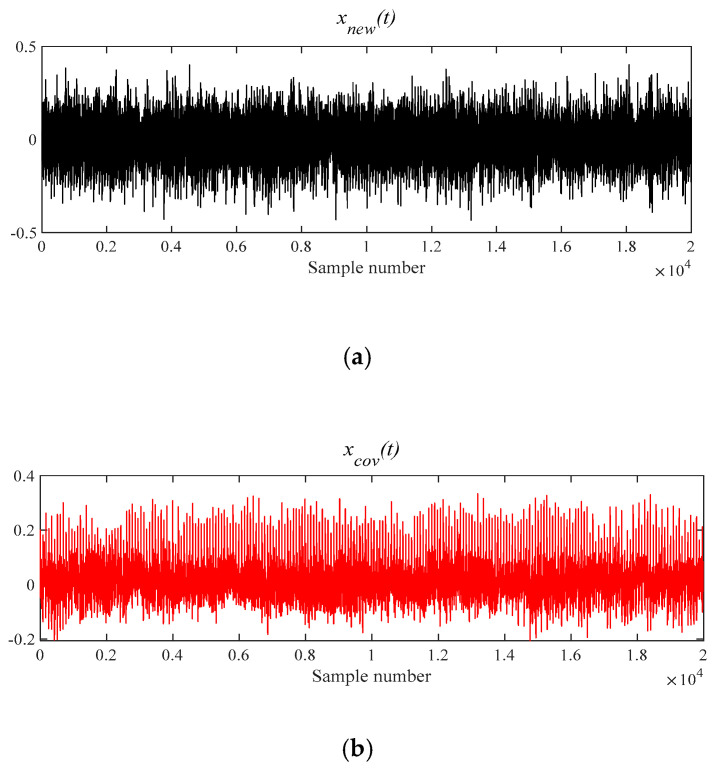
The signal processing by MOMEDA method. (**a**) Reconstruction signal *x__new_* (*t*); (**b**) Periodic pulse signal *x__cov_*(*t*).

**Figure 27 entropy-21-00331-f027:**
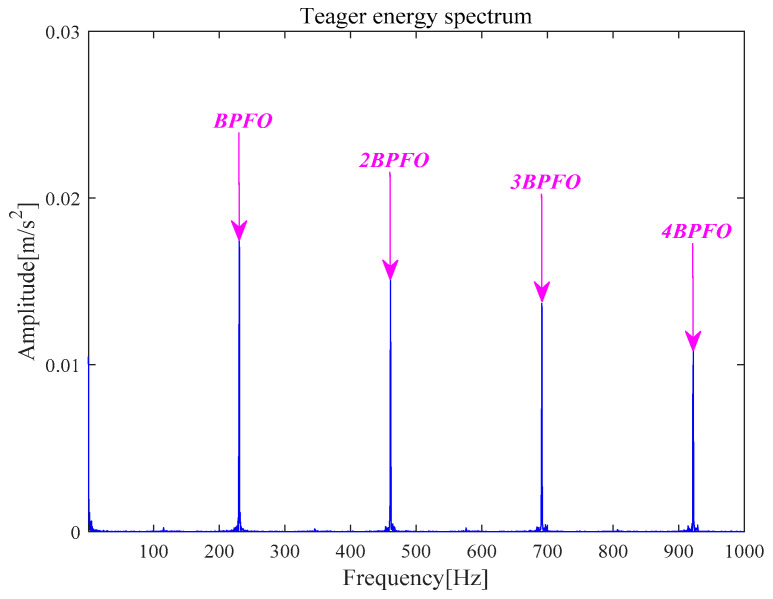
Teager energy spectrum.

**Figure 28 entropy-21-00331-f028:**
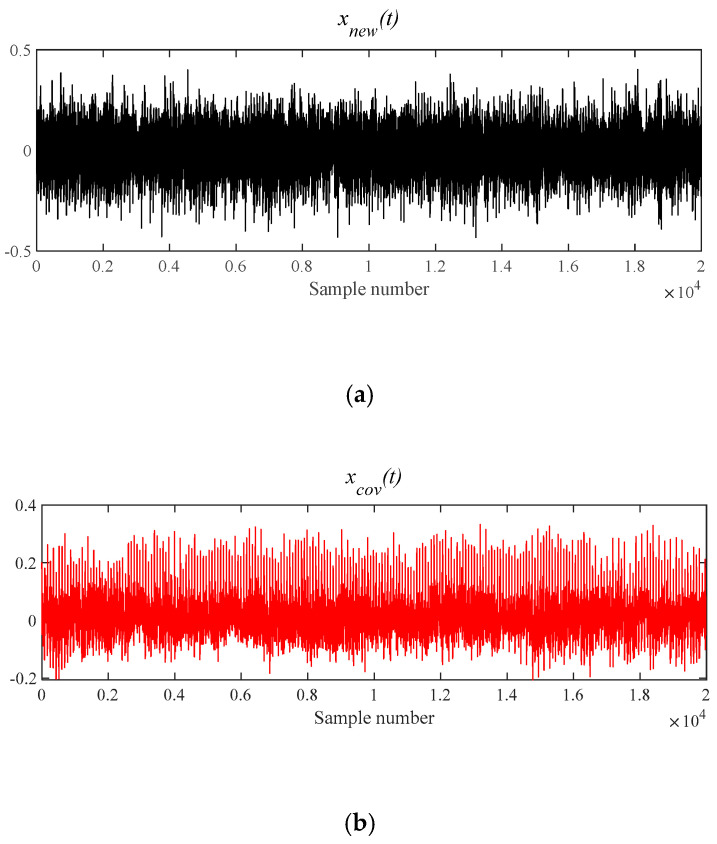
The signal processing by MOMEDA method. (**a**) Reconstruction signal *x__new_* (*t*); (**b**) Periodic pulse signal *x__cov_*(*t*).

**Figure 29 entropy-21-00331-f029:**
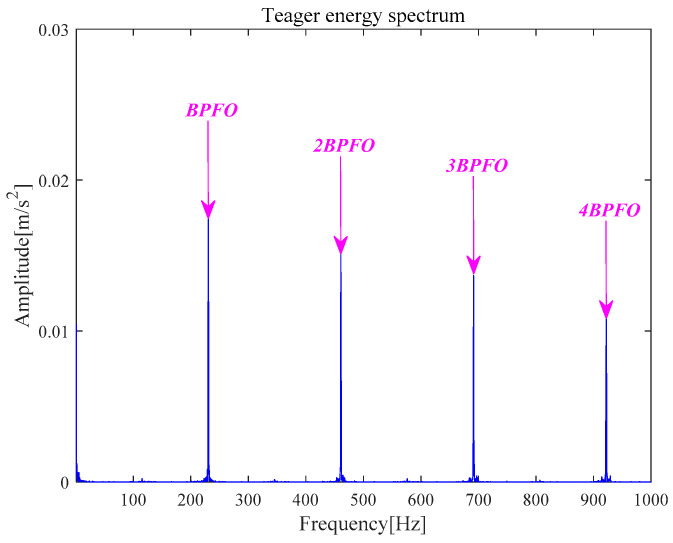
Teager energy spectrum.

**Figure 30 entropy-21-00331-f030:**
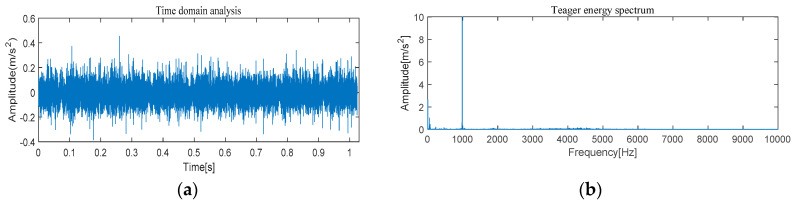
Original normal signal time–frequency analysis. (**a**) Time domain analysis. (**b**) Frequency domain analysis.

**Figure 31 entropy-21-00331-f031:**
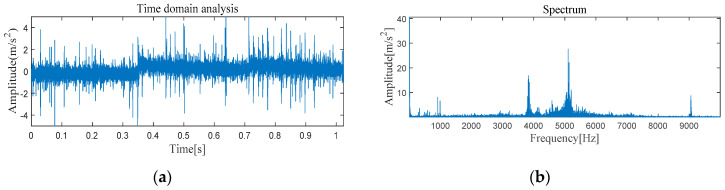
Original fault signal time–frequency analysis. (**a**) Time domain analysis. (**b**) Frequency domain analysis.

**Figure 32 entropy-21-00331-f032:**
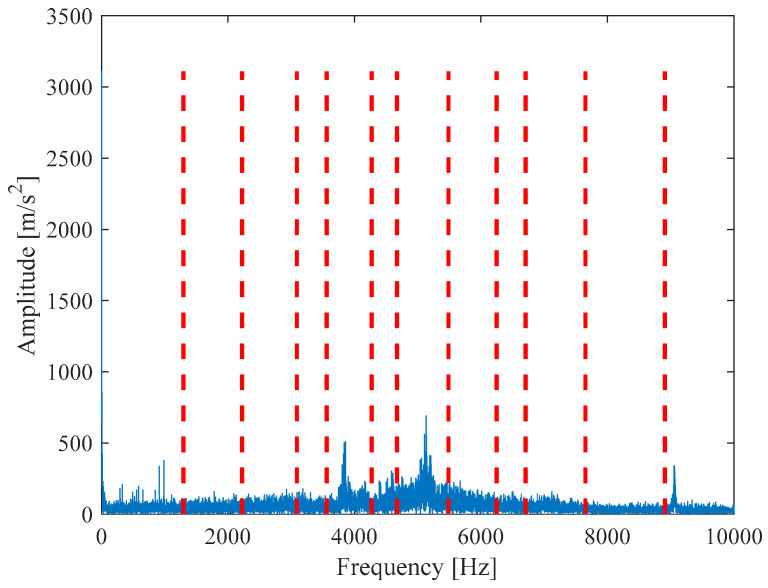
Scale spectrum segmentation boundary.

**Figure 33 entropy-21-00331-f033:**
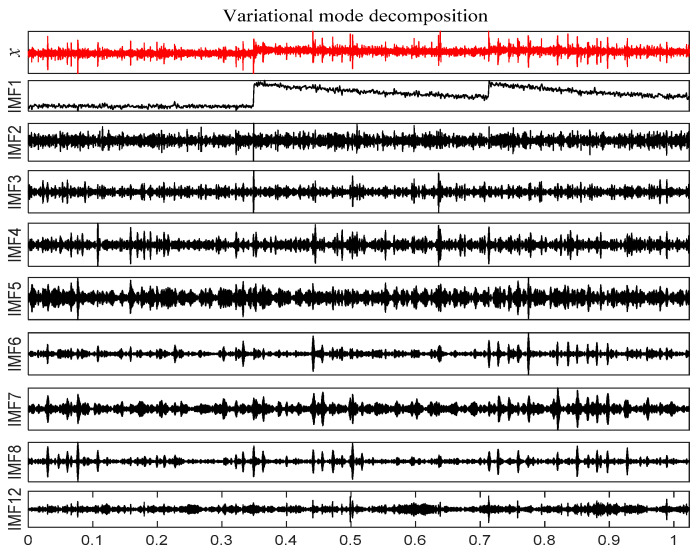
IMF effective component time domain waveform.

**Figure 34 entropy-21-00331-f034:**
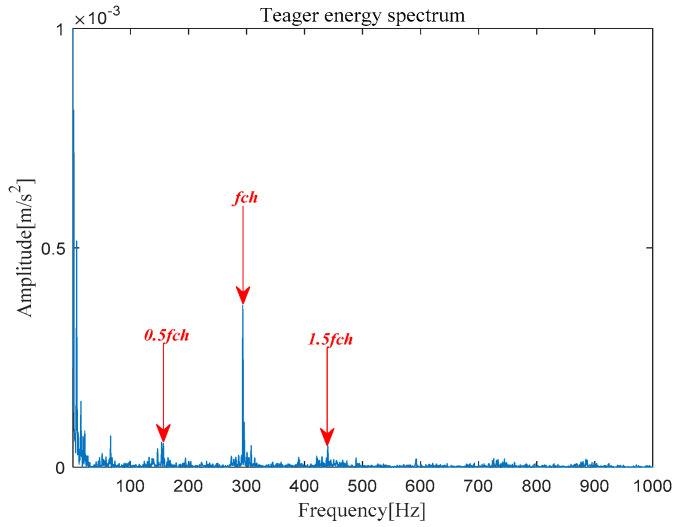
Teager energy spectrum.

**Figure 35 entropy-21-00331-f035:**
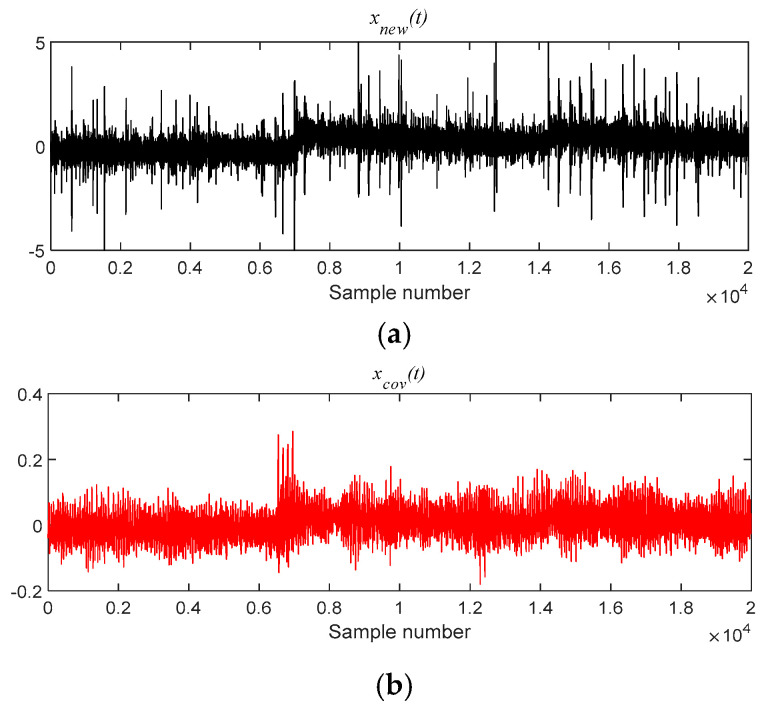
The signal processing by MOMEDA method. (**a**) Reconstruction signal *x__new_* (*t*); (**b**) Periodic pulse signal *x__cov_*(*t*).

**Figure 36 entropy-21-00331-f036:**
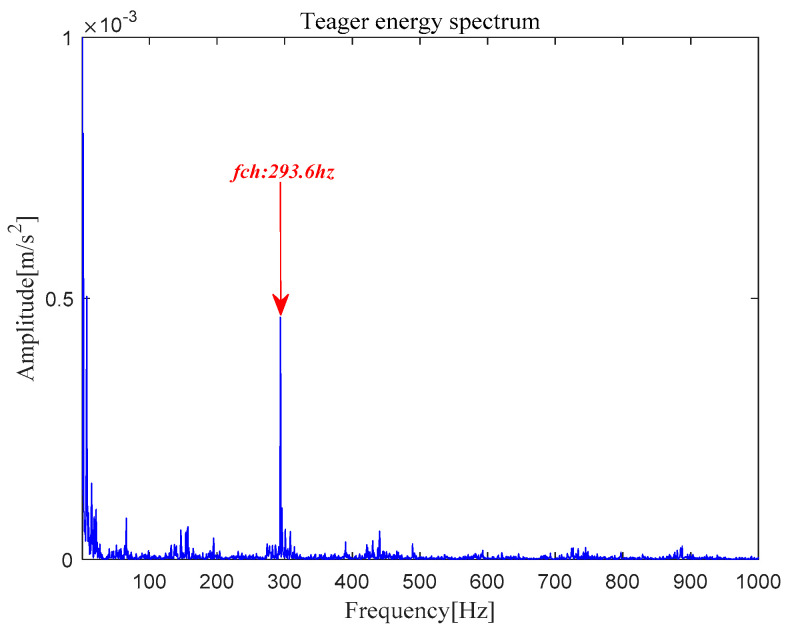
Teager energy spectrum.

**Table 1 entropy-21-00331-t001:** Bearing basic parameters.

Model	Rolling Element Number (*Z*)	Inner Diameter (inches)	Outer Diameter (inches)	Contact Angle (*θ*)	Rolling Element Diameter *d* (inches)	Pitch Circle Diameter *D* (inches)	Speed (rpm)
6205-2RSJEMSKF	9	0.9843	2.0472	0°	0.3126	1.537	1797

**Table 2 entropy-21-00331-t002:** Energy ratio of each IMF component.

Decomposed Component	IMF1	IMF2	IMF3	IMF4	IMF5	IMF6	IMF7	IMF8	IMF9	IMF10
*E*(*t*)	/	0.01	0.05	0.12	0.11	0.25	0.29	0.16	/	0.01

‘/’ represents that the energy ratio *E*(*t*) of IMF component is close to infinity or negligible.

**Table 3 entropy-21-00331-t003:** Energy ratio of each IMF component.

**Decomposed Component**	**IMF1**	**IMF2**	**IMF3**	**IMF4**	**IMF5**	**IMF6**
*E*(*t*)	0.01	0.06	0.12	0.03	0.16	0.21
**Decomposed Component**	**IMF7**	**IMF8**	**IMF9**	**IMF10**	**IMF11**	
*E*(*t*)	0.09	0.11	0.08	0.02	0.01	

**Table 4 entropy-21-00331-t004:** Bearing basic parameters.

Rolling Element Number (Z)	Contact Angle (*θ*)	Rolling Element Diameter *d* (mm)	Pitch Diameter *D* (mm)	Rotational Speed (rpm)
16	15.17°	0.331	2.815	2000

**Table 5 entropy-21-00331-t005:** Energy ratio of each IMF component.

**Decomposed Component**	**IMF1**	**IMF2**	**IMF3**	**IMF4**	**IMF5**	**IMF6**
*E*(*t*)	0.02	0.07	0.05	0.10	0.09	0.33
**Decomposed Component**	**IMF7**	**IMF8**	**IMF9**	**IMF10**	**IMF11**	**IMF12**
*E*(*t*)	0.21	0.09	0.02	0.01	0.01	/

‘/’ represents that the energy ratio *E*(*t*) of IMF component is close to infinity or negligible.

**Table 6 entropy-21-00331-t006:** Energy ratio of each IMF component.

**Decomposed Component**	**IMF1**	**IMF2**	**IMF3**	**IMF4**	**IMF5**	**IMF6**
*E*(*t*)	0.38	0.03	0.03	0.04	0.09	0.07
**Decomposed Component**	**IMF7**	**IMF8**	**IMF9**	**IMF10**	**IMF11**	**IMF12**
*E*(*t*)	0.18	0.06	0.03	0.03	0.02	0.04

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
