# Peer review of "MVMD-MOMEDA-TEO Model and Its Application in Feature Extraction for Rolling Bearings"

_entropy, 2019, doi:10.3390/e21040331_

Round 1

Reviewer 1 Report

With regards to technical soundness, relevance and academic writing style, the submitted manuscript that primarily deals with feature extraction for rolling element bearings is quite good. However, I feel that the review of existing literature especially with reference to practically applied techniques is quite shallow and weak. There was not reference to conventional techniques for extracting bearing fault features such as kurtosis, Hilbert-Huang transform, higher order spectra, combination of FFT and coherence for feature extraction and noise suppression during faults diagnosis. It will be very useful and a significant means of enriching the background review section within this article to highlight the trend especially with regards to industry-dominant techniques, their strengths, limitations and what benefits your approach has over them. In the current form of the article, you mainly focused on the limitations of wavelet analysis but this isn't the only denoising approach. I recommend that you highlight past useful studies including the following, so as to boost the robustness of the review section:

Rai VK, Mohanty AR. Bearing fault diagnosis using FFT of intrinsic mode functions in Hilbert–Huang transform. Mechanical Systems and Signal Processing. 2007 Aug 1;21(6):2607-15.

Yang DM, Stronach AF, MacConnell P, Penman J. Third-order spectral techniques for the diagnosis of motor bearing condition using artificial neural networks. Mechanical systems and signal processing. 2002 Mar 1;16(2-3):391-411.

Yunusa-Kaltungo A, Sinha JK. Faults diagnosis in rotating machines using higher order spectra. InASME turbo expo 2014: turbine technical conference and exposition 2014 Jun 16 (pp. V07AT31A002-V07AT31A002). American Society of Mechanical Engineers.

Sawalhi N, Randall RB, Endo H. The enhancement of fault detection and diagnosis in rolling element bearings using minimum entropy deconvolution combined with spectral kurtosis. Mechanical Systems and Signal Processing. 2007 Aug 1;21(6):2616-33.

Luwei KC, Yunusa-Kaltungo A, Sha’aban YA. Integrated Fault Detection Framework for Classifying Rotating Machine Faults Using Frequency Domain Data Fusion and Artificial Neural Networks. Machines. 2018 Nov 20;6(4):59.

Stack JR, Harley RG, Habetler TG. An amplitude modulation detector for fault diagnosis in rolling element bearings. IEEE Transactions on Industrial Electronics. 2004 Oct;51(5):1097-102.

Secondly, I observed that the authors used a sampling frequency of 12kHz and 2048 data points which implies a frequency resolution of approximately 5.9Hz. Can the authors provide explanations on how the adequacy of these signal processing parameters were adjudged adequate and were able to yield the plots/results displayed in this study.

Finally, please perform a detailed proofreading of the manuscript especially introduction, so as to further eliminate residual typographical and grammatical errors. If all these can be addressed, then the article has a good potential for publication.

Author Response

Response to Reviewer 1 Comments

With regards to technical soundness, relevance and academic writing style, the submitted manuscript that primarily deals with feature extraction for rolling element bearings is quite good.

Point 1:The review of existing literature especially with reference to practically applied techniques is quite shallow and weak. There was not reference to conventional techniques for extracting bearing fault features such as kurtosis, Hilbert-Huang transform, higher order spectra, combination of FFT and coherence for feature extraction and noise suppression during faults diagnosis. It will be very useful and a significant means of enriching the background review section within this article to highlight the trend especially with regards to industry-dominant techniques, their strengths, limitations and what benefits your approach has over them. In the current form of the article, you mainly focused on the limitations of wavelet analysis but this isn't the only denoising approach.

Response 1:Thank you for your valuable comment and guidance. The review of existing literature is modified and described as follows.

To extract bearing fault features, Hilbert-Huang transform (HHT) [4], kurtosis [5], high-order spectrum [6-7], Wavelet Transform (WT) [8], Empirical Mode Decomposition (EMD) [9], Local Mean Decomposition (LMD) [10], and other methods have been proposed and achieved some results. Nevertheless, these methods have their own limitations. HHT has some problems, such as unexplained negative frequency and energy leakage caused by endpoint effect. High-order spectrum has good effect in signal processing and fault feature extraction of nonlinear systems, but its computational complexity is larger than other algorithms. WT needs to preset the wavelet basis and decomposition scale, and the result is a fixed frequency band signal without self-adaptability. Although EMD and LMD methods can adaptively decompose complex signals into a series of components, there are still some theoretical problems, such as envelope, mode aliasing, endpoint effect, IMF criterion.

Combining the idea of solving modal bandwidth with constrained optimization, Dragomiretskiy [11] proposed variational mode decomposition (VMD). This method used an iterative method to search the optimal solution of the variational model, then determined the central frequency and bandwidth of each component, so that the effective separation of signal frequency domain can be realized adaptively. Compared with EMD and LMD, there is no mode mixing and endpoint effect. Because of the above advantages, VMD method has been widely used in rolling bearing fault feature extraction since it was proposed [12-14]. However, there are two limitations for VMD: (1) the number K of decomposition components must be given beforehand; (2) the selection of control parameters for VMD lacks theoretical basis. For non-linear and unsteady signals, the preset the number K of decomposition modes may lead to information loss or over-decomposition, which affects the performance of subsequent feature extraction [15]. Therefore, how to quickly and adaptively determine the decomposition mode number K of VMD for improving the speed of signal processing is one of the urgent problems to be solved. Therefore, the modified variational mode decomposition (MVMD) method proposed in [16] is introduced to determine the decomposition mode number K of VMD rapidly and accurately.

A certain amount of noise still remains in each IMF component obtained by decomposition method without exception. To improve the accuracy of fault feature extraction, it is necessary to further enhance the periodic effective pulse of the fault vibration signal such as the bearings, and denoise the reconstructed signal after decomposition. Wiggins [17] proposed minimum entropy deconvolution (MED). H. Endoet et al. firstly applied MED to fault detection of rotating machinery [18]. For reducing the influence of noise and extract the fault feature information of rolling bearings accurately, N. Sawalhi[19] presented an algorithm for enhancing the surveillance capability of Spectral kurtosis by using the minimum entropy deconvolution (MED) technique. The MED technique effectively deconvolved the effect of the transmission path and clarifies the impulses, even where they are not separated in the original signal. Ren Xueping[20] proposed a fault characteristics extraction method of rolling bearings based on the combination of VMD and MED. The fault signal of rolling bearing is decomposed by VMD method, and then the reconstructed signal is processed by MED denoising. The fault feature information is extracted from envelope spectrum accurately. However, the MED method is not only complex in operation, but also not necessarily the global optimal filter. Moreover, the MED method is only suitable for single impulse signals. Wang Jianguo [21] and Xia Junzhong [22] proposed a bearing fault diagnosis method based on the combination of VMD and maximum correlation kurtosis deconvolution (MCKD). After VMD decomposition of the fault signal, MCKD was used to reduce the noise of each IMF component and highlight the fault impact component to obtain accurate bearing fault characteristic frequency. But the MCKD method needs to preset the core parameters such as the fault period, which is inconsistent with the reality. Because the fault period may not be known or calculated in advance. To solve the above mentioned problems, McDonald et al.[23] proposed a multipoint optimal minimum entropy deconvolution adjusted(MOMEDA) method, defined the target vector and D-norm, and effectively solved the design problem of the optimal filter. The MOMEDA algorithm does not need to preset the fault cycle, nor does it need to iterate. The impulse component can be accurately extracted by using the multipoint kurtosis spectrum.

Point 2:I observed that the authors used a sampling frequency of 12kHz and 2048 data points which implies a frequency resolution of approximately 5.9Hz. Can the authors provide explanations on how the adequacy of these signal processing parameters were adjudged adequate and were able to yield the plots/results displayed in this study.

Response 2: Thank you for your valuable comment and guidance. In view of the above problems, the following two explanations are given.

(1) In order to distinguish the outer race fault frequency from the bearing rotation frequency, it is necessary to ensure that the frequency resolution(5.9Hz) is less than the difference between the outer race fault frequency and the bearing rotation frequency (BPFO-fr=77.41Hz).

(2) In FFT, the number of data points is usually 2N.

(3) At the same time, if there are too many data points, the algorithm will be time consuming.

In summary, the sample number N=2048 is selected to verify the proposed MVMD-MOMEDA-TEO method.

Point 3: Please perform a detailed proofreading of the manuscript especially introduction, so as to further eliminate residual typographical and grammatical errors.

Response 3: Thank you for your valuable comment and guidance. We have checked and corrected the manuscript carefully, we hope now the detailed proofreading of the manuscript is suitable for publishing.

Reviewer 2 Report

The shafts of various rotating machines are supported by rolling bearings. Though the friction in rolling bearings is low in comparison with hydrodynamic bearings, their reliability is also diminished because of the high pressure concentrated contacts between the rolling elements and races.  The diagnosis of rolling bearings is a crucial issue, but it cannot usually be realized by Fourier transform or time analysis of vibrational signals, as the signal generated by a defect rolling bearing is non-stationary. Also, a lot of noise is accompanying the running of rolling bearings, noise derived from the environment, geometrical form deviation, mounting errors, and fitting tolerances of different parts of the machine.

Taking into account the above, the authors of the current paper proposed a new combined technique to predict the faults of inner and outer races of ball bearings. The new predictive model combines the modified variational mode decomposition (MVMD) and multipoint optimal minimum entropy deconvolution adjusted (MOMEDA). Finally, the teager energy operator demodulation (TEO) is employed to calculate Teager energy spectrum.

The model is applied to the processing of vibrational signals of faulty bearings available from databases (CWRU and NASA). A comparison of results of the present model ( MVMD - MOMEDA - TEO) against those obtained from two other models (MVMD - TEO and MOMEDA - TEO) is also presented in the article.

Personally I appreciate the huge quantity of work realized by the authors, but I consider that the results are below expectations. 

Consequently, to increase the quality of the paper, I have some suggestions and comments for the authors.

Major revision

1. The paper presents multiple similar results acquired from the previously mentioned different models. As there are no differences in findings, try to concentrate the presentation, even omitting the similar graphical results.

2. The pictures are not very well arranged, so it is difficult to localize them in the paper. There is also a missing Figure 20. Rearrange all the Figures from Figure 5 to Figure 19, as in the present form they are scarcely identifiable.

3. For the NASA rolling bearing you computed the rotation frequency around 33.33 Hz, but in the Teager energy spectrum of the healthy bearing (Figure 22b) we find a very high peak just around 1,000 Hz. Normally we should find the rotational frequency. How can you explain this?!

4. It is well known that dedicated industrial diagnosing devices for rolling bearings implement only the Fast Fourier Transform (FFT). Instead of presenting similar results from the 3 different methods, I advise the authors to present the (FFT) analysis of the raw signals, or rather the FFT analysis of the signals' envelope . In such a way they could emphasize the improvements brought by their new combined model. 

5. In sections 2.2 and 2.3, MOMEDA and TEO should be detailed.

Minor revision

1. The nomenclature of fch should be mentioned below equation (2), and also details on its computation formula. Also, provide the values of Î¼ factor.

2.  The diagrams are not clear due to emboss and engraves of some text. Please use normal text.

3. Section 4.2, line 278: the mentioned Figure 19 is in fact Figure 21.

4. Please try to minimize the repetitive assertions : "It can be judged that the bearing is abnormal", as it is difficult to diagnose a bearing by simple observation of time variation of acquired signal. In my opinion, it can be observed just more noise.

5. Figures 22 to 36 should be rearranged, especially the twin figures presented in a single column.

Author Response

Response to Reviewer 2 Comments

Point 1: The paper presents multiple similar results acquired from the previously mentioned different models. As there are no differences in findings, try to concentrate the presentation, even omitting the similar graphical results.

Response 1: Thank you for your valuable comment and guidance. In view of the above problems, the following two explanations are given.

(1) At the beginning of this paper, although the experimental results are similar to those of the proposed method may appear in the comparison method, the comparative experiment using two datasets such as CWRU and NASA is to verify the applicability and extensiveness of the proposed method.

(2) As shown in Fig. 34 and Fig. 36, it is well known that the vibration signals of NASA bearing inner race contain lots of noise components. It can be seen from the figures, the proposed method has better adaptability.

Fig.34 Teager energy spectrum(MVMD-MOMEDA-TEO)    Fig. 36 Teager energy spectrum(MOMEDA-TEO)

Point 2: The pictures are not very well arranged, so it is difficult to localize them in the paper. There is also a missing Figure 20. Rearrange all the Figures from Figure 5 to Figure 19, as in the present form they are scarcely identifiable.

Response 2: Thank you for your careful examination of our paper. We have rearrange and corrected the manuscript carefully, we hope the present form of the manuscript is suitable for publishing.

Point 3: For the NASA rolling bearing you computed the rotation frequency around 33.33 Hz, but in the Teager energy spectrum of the healthy bearing (Figure 22b) we find a very high peak just around 1,000 Hz. Normally we should find the rotational frequency. How can you explain this?

Response 3: Thank you for your valuable comment and question. It is well known that the vibration signal of rolling bearings collected by NASA is more complex than that of CWRU, and the rotational frequency is not obvious,and some unknown irregular prominent components often appear. It’s shown from Fig.1 that the frequency domain analysis of the normal vibration signal. Fig.3 shows the Teager energy spectrum obtained by using the proposed method to analyze the normal signal. In order to make a better comparison between the traditional frequency domain analysis and the proposed method, frequency range is shorten to observe the frequency characteristics. It can be seen from Fig.3 that frequency 33.57 Hz approach theoretical rotational frequency. Compared with Fig.2, the amplitude is more obvious. It can be proved that better feature extraction results can be obtained by using the proposed method.

Fig.1 original nomal signal frequency analysis

Fig.2 original nomal signal frequency analysis      Fig.3 Teager energy specertrum (MVMD-MOMEDA)

Point 4: It is well known that dedicated industrial diagnosing devices for rolling bearings implement only the Fast Fourier Transform (FFT). Instead of presenting similar results from the 3 different methods, I advise the authors to present the (FFT) analysis of the raw signals, or rather the FFT analysis of the signals' envelope. In such a way they could emphasize the improvements brought by their new combined model.

Response 4: Thank you for your valuable comment and guidance. In this paper, Teager energy spectrum is used to represent the relevant information of the original signal in frequency domain. Because Teager energy operator can enhance the signal’s transient characteristics, high time resolution, low computational complexity, high algorithm efficiency and outstanding advantages suitable for detecting impulse components in the signal.

Point 5: In sections 2.2 and 2.3, MOMEDA and TEO should be detailed.

Response 5: Thank you for your valuable comment and guidance. MOMEDA and TEO methods are modified and described as follows.

2.2 MOMEDA method

The purpose of MOMEDA algorithm is to find the optimal FIR filter in a non-iterative way and reconstruct the vibration and shock signal y. The deconvolution process is as followed:

                                                       (3)

Among them, , according to the characteristics of periodic impulse signal, the method introduces multi-point D-norm :

                                                     (4)

                                  (5)         

In the Eq.(4)the constant vector  is used to determine the position and weight of the target impact component. The optimal filter f is obtained by solving the maximum of the multi-point D-norm, and the deconvolution process also obtains the optimal solution.

MOMEDA uses multipoint kurtosis (MKurt) to determine the maximum position of the pulse.

                          (6)

Referring to the parameter selection rule in reference [25], the length range of the filter is 20~500, and the periodic parameters should cover the frequency range analyzed. This article takes the default value ,and carries on the analysis.

2.3 TEO Demodulation Principle

For the definition of continuous signal x(t):TEO demodulation j[x(t)], reference [26]:

j[x(t)]=[]2 - x(t)                                         (7)

The  and is the first and second order differential of x(t) to time t respectively.

For discrete signal x(n), differential is used instead of differential. Then j[x(n)] is defined as:

j[x(n)]=[x(n)]2-x(n-1)x(n+1)                                     (8)

It is known from Eq.(8) that for discrete signal x(n), TEO only needs three sample data to calculate the signal source energy at any time n. For the IMF component of the vibration signal, the TEO demodulation envelope signal j[PF] of the IMF component can be calculated according to Eq.(8), and the subsequent Fourier spectrum analysis is performed by using j[PF] instead of the original signal x(n). The spectral characteristics of the vibration signal are extracted to determine the fault.

Point 6: The nomenclature of fch should be mentioned below equation (2), and also details on its computation formula. Also, provide the values of μ factor.

Response 6: Thank you for your valuable comment and guidance. MOMEDA and TEO methods are modified and described as follows.

                                                                 (2)

In formula (2), there is no strict restriction on the selection of values, and the recommended range of values is [2-4]. At the same time, when the signal is modulated by fault characteristic frequency fch and noise pollution, the small change of scale parameters has no obvious influence on the final analysis results. Therefore, the scale parameter is chosen as [16].

Point 7: The diagrams are not clear due to emboss and engraves of some text. Please use normal text.

Response 7: Thank you for your valuable comment and guidance. We have checked and corrected the diagram carefully, we hope now the detailed diagram of the manuscript is suitable for publishing.

Point 8: Section 4.2, line 278: the mentioned Figure 19 is in fact Figure 21.

Response 8: Thank you for your valuable comment and guidance. We have checked and corrected the diagram and description.

Point 9: Please try to minimize the repetitive assertions : "It can be judged that the bearing is abnormal", as it is difficult to diagnose a bearing by simple observation of time variation of acquired signal. In my opinion, it can be observed just more noise.

Response 9: Thank you for your valuable comment and guidance. We have checked and corrected the manuscript carefully.

Round 2

Reviewer 1 Report

Authors have adequately addressed all of my initial concerns.

Reviewer 2 Report

The authors responded to all my questions and took into account all my suggestions. The paper is very clear now as it was much improved.